# Rapid and efficient hydrogen clathrate hydrate formation in confined nanospace

Judit Farrando-Perez[1,3], Rafael Balderas-Xicohtencatl[2,3], Yongqiang Cheng[2], Luke Daemen[2], Carlos Cuadrado-Collados[1], Manuel Martinez-Escandell [1], Anibal J. Ramirez-Cuesta [2] ✉ & Joaquin Silvestre-Albero [1] ✉

Clathrate hydrates are crystalline solids characterized by their ability to accommodate large quantities of guest molecules. Although $CH_4$ and $CO_2$ are the traditional guests found in natural systems, incorporating smaller molecules (e.g., $H_2$) is challenging due to the need to apply higher pressures to stabilize the hydrogen-bonded network. Another critical limitation of hydrates is the slow nucleation and growth kinetics. Here, we show that specially designed activated carbon materials can surpass these obstacles by acting as nanoreactors promoting the nucleation and growth of $H_2$ hydrates. The confinement effects in the inner cavities promote the massive growth of hydrogen hydrates at moderate temperatures, using pure water, with extremely fast kinetics and much lower pressures than the bulk system.

Clathrate hydrates are one of the most exciting crystalline structures in nature. Under favorable pressure and temperature conditions, water molecules can be arranged in a 3D hydrogen-bonded network, giving rise to void spaces or cages in the size range of many gas molecules[1,2]. Overall, clathrate hydrates can be considered the main reservoir on Earth for gas storage. Natural clathrate hydrates contain methane preferentially as a guest, although carbon dioxide hydrates have also been identified. Despite their abundance, their formation was deemed restricted to molecules above 0.33–0.36 nm due to the necessity to stabilize the 3D network through intracrystalline non-bonding interactions.

Hydrogen clathrate hydrates could be potential reservoirs for hydrogen storage with a maximum capacity (considering the full occupation of cages) of ca. 5.0 wt.% and 46.7 g/L[3–6]. Previous studies show that hydrogen molecules can be enclathrated in the cages of hydrogen-bonded water molecules with an sII crystalline structure[4]. However, at this stage, there were two critical obstacles to be solved for a potential application of the clathrate hydrates in a hydrogen-based economy: (i) the high pressures needed to promote the nucleation and growth of these $H_2$ hydrates and (ii) the slow nucleation kinetics, associated with the limited gas-liquid interface. Indeed, due to the small kinetic diameter of hydrogen (0.272 nm) and the low mass

that creates quantum effects, pressures above 200 MPa using a diamond anvil cell (DAC) at 249 K were required to promote the formation of these crystalline structures on a small scale[4].

In this letter, we demonstrate the formation of hydrogen clathrates at lower pressures than the bulk system (30% pressure reduction), with extremely fast kinetics (within minutes) and almost complete conversion. Firstly, we design a porous substrate (activated carbon) that facilitates the massive growth of hydrogen clathrates without incorporating additives. Secondly, we take advantage of the percolation paths for hydrogen to reach the nucleation sites to improve nucleation and growth kinetics. Based on these premises, we propose a novel strategy for efficient synthesis of $H_2$ clathrate hydrates avoiding traditional technological obstacles. Inspired by nature, where methane hydrates are grown in pores and voids of natural sediments, we propose using specially designed high-surface-area carbon materials as nanoreactors to promote the nucleation and growth of $H_2$ hydrates. Taking advantage of the nanoconfinement effects, we grew massive $H_2$ hydrates using pure water, with a nearly complete water-to-hydrate conversion, within minutes and significantly at lower pressures than the bulk system. This development could be the breakthrough that will make hydrogen clathrates a viable hydrogen storage technology.

[1]Laboratorio de Materiales Avanzados, Departamento de Química Inorgánica-Instituto Universitario de Materiales, Universidad de Alicante, Alicante, Spain. [2]Spallation Neutron Source, Oak Ridge National Laboratory, Oak Ridge, TN 37831, USA. [3]These authors contributed equally: Judit Farrando-Perez, Rafael Balderas-Xicohtencatl. ✉e-mail: ramirezcueaj@ornl.gov; joaquin.silvestre@ua.es

## Results

The Petroleum Pitch Activated Carbon (PPAC) sample is an optimized activated carbon combining a tailored porous structure with proper surface chemistry. It was prepared from petroleum pitch residue using KOH as activating agent. A large apparent surface area characterizes the synthesized material ($S_{BET}$ ~3690 m²/g), with widely developed microporosity ($V_{micro}$ ~1.06 cm³/g) and mesoporosity ($V_{meso}$ ~1.90 cm³/g). Figure 1 shows the textural and morphological characteristics of the synthesized carbon. Also, the activated carbon sample exhibits wide mesopores and macropores associated with the voids left by KOH after the activation step (see Hg porosimetry data in Fig. 1c).

One of the main characteristics of sample PPAC is its large adsorption capacity for $H_2O$ at high relative pressures (saturation conditions), up to ca. 1.8 $g_{H2O}/g$[7]. This capacity is due to the presence of a highly developed porous structure and oxygen surface groups (ca. 4 at.%)—see Fig. 1, Supplementary Fig. 1 and Supplementary Table 1 for SEM, TEM, and XPS measurements.

Based on our previous experience with confined gas hydrates[7,8], the PPAC sample was impregnated dropwise with $D_2O$ ($D_2O$-PPAC) up to oversaturation ($R_w$ = 4.1 $g_{H2O}/g$) before the high-pressure experiments with hydrogen. The main goal when using oversaturation conditions was to fill with $D_2O$ not only micro and mesopores (1.8 $g_{H2O}/g$) but also macrocavities left after removing the activating agent (KOH). Around 1 g of the $D_2O$-PPAC was loaded into a CuBe cell and connected to an online high-pressure gas system. Experiments were performed at the Spallation Neutron Source at Oak Ridge National Lab using the VISION neutron spectrometer. Inelastic neutron scattering (INS) is highly sensitive to the dynamics of hydrogen in atomic and molecular form and has no selection rules[9,10]. The neutron experiments used $D_2O$ to minimize the incoherent scattering signal from the water framework while simultaneously following the diffraction pattern of the structure.

The INS spectral signatures in VISION are in the low energy transfer region (0-40 meV), as shown in Fig. 2a and b, either PPAC or $D_2O$-PPAC, the signal in the spectra is entirely flat. The CuBe cell (ca. 1 cm³) is loaded at 350 K with high-pressure hydrogen gas. Once the pressure is stabilized, the sample is cooled down to 5 K (Supplementary Fig. 2). Excess hydrogen pressure is released at 160 K during the cooling process, i.e., sample cell is evacuated down to 0.1 MPa. Once the sample reaches 5 K, the INS spectra are recorded. It is essential to

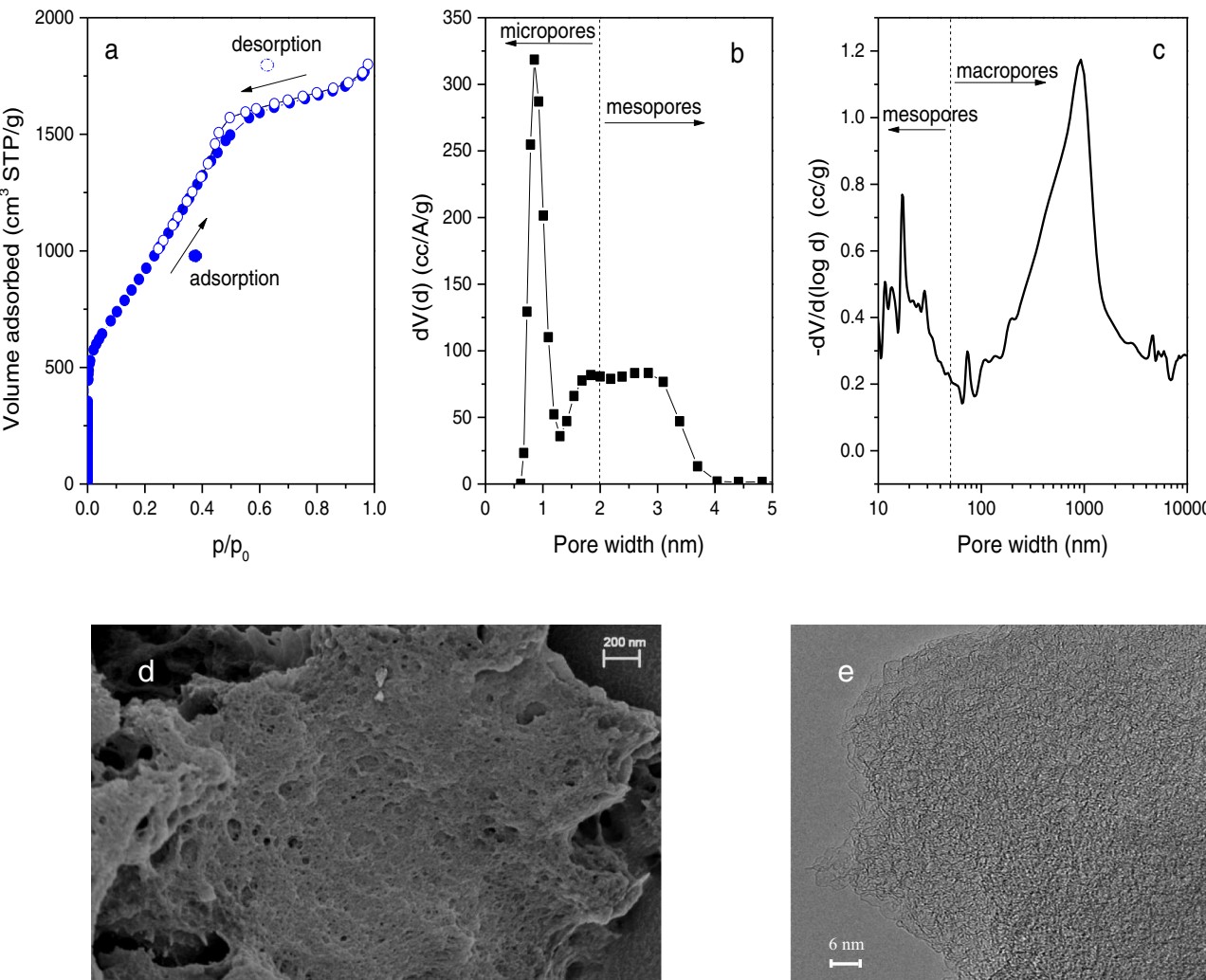

**Fig. 1 | Textural and morphological characteristics of the synthesized activated carbon PPAC. a** $N_2$ adsorption (full circles)/desorption (empty circles) isotherm measured in a manometric equipment at 77 K. The $N_2$ isotherm demonstrates the presence of a widely developed porous structure combining micro- and mesopores. **b** Pore size distribution (PSD) obtained after application of the quenched solid density functional method—QSDFT (slit-shaped, equilibrium model). The PSD profile confirms the presence of narrow micropores, around 1 nm, and some mesopores in the region around 2–3 nm. **c** Hg porosimetry profile shows the presence of large mesopores and macropores in sample PPAC. These large cavities are due to the voids created after the removal of the activating agent. **d** FESEM and **e** TEM images of the petroleum-pitch activated carbon—PPAC (micropores, mesopores and macropores can be appreciated). Source data are provided as a Source Data file.

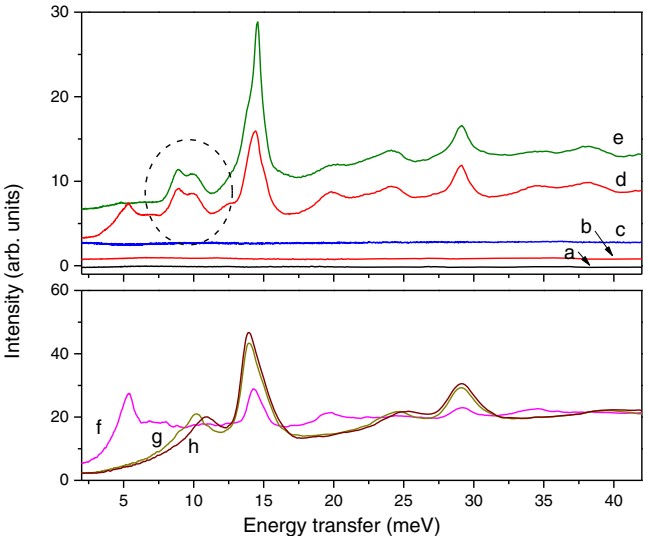

**Fig. 2 | Inelastic neutron scattering analysis at increasing H₂ pressures.** (Upper panel) Background subtracted INS patterns at 5 K of (**a**) sample PPAC original, (**b**) $D_2O$-PPAC, and after pressurizing the $D_2O$-PPAC with hydrogen at (**c**) 100, (**d**) 135, and (**e**) 200 MPa. INS patterns show the appearance of new peaks at hydrogen pressures of 135 MPa and above attributed unambiguously to the formation of hydrogen hydrates. (Lower panel) INS patterns at 5 K of normal H₂ at (**f**) atmospheric pressure, (**g**) 170 MPa, and (**h**) 240 MPa. INS patterns show that solid hydrogen exhibits a characteristic contribution at low energy transfer (*ca.* 10 meV under high pressure conditions). Interestingly, the doublet observed for enclathrated hydrogen (**d**) fits with the signal of solid hydrogen under pressure (**g**). Source data are provided as a Source Data file.

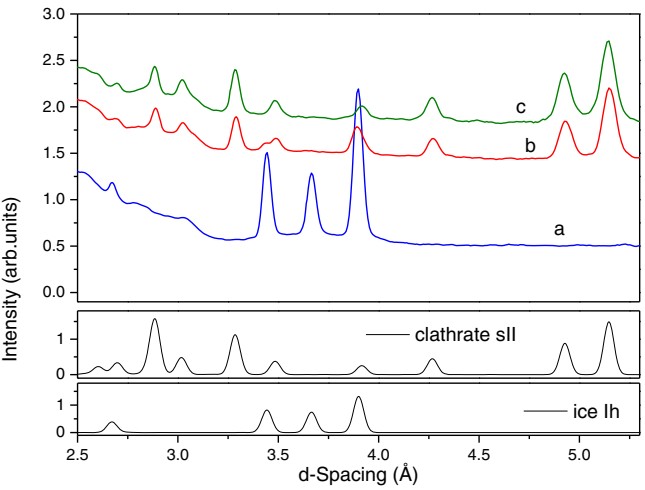

**Fig. 3 | Neutron diffraction (ND) analysis at increasing H₂ pressures.** Neutron diffraction patterns obtained at 5 K for the $D_2O$-PPAC pressurized with normal hydrogen at (**a**) 100, (**b**) 135, and (**c**) 200 MPa. Theoretical patterns for hexagonal ice (Ih) and sII structure in gas hydrates are included for comparison. ND patterns confirm the formation of hydrogen hydrates with the traditional sII structure at hydrogen pressures of 135 MPa and above. Lower pressures are not sufficient to promote hydrogen enclathration, hexagonal ice being the main component in the $D_2O$-PPAC system. Source data are provided as a Source Data file.

highlight that 5 K is the conventional temperature used when measuring INS to get high resolution spectroscopic vibrational signals, i.e., a minimum in the Debye-Waller factor. The INS differ depending on the final pressure applied upon pressurization with hydrogen. At 100 MPa, the spectrum does not show any new features. However, when the pressure is 135 MPa or higher, the INS spectra are dominated by an intense peak around 14.7 meV and a doublet with maxima at 8.95 meV and 10.2 meV (Fig. 2d and e)[11]. The peak at 14.7 meV is the rotational line of parahydrogen, while the doublet contribution is assigned unambiguously to orthohydrogen vibrational transitions. Previous studies with binary THF-H₂ hydrates attributed the signal between 8 and 10 meV to orthohydrogen, more specifically, to the combination of the elastic $J = 1 \rightarrow 1$ rotational transition with the fundamental rattling transition of the center-of-mass motion (see Supplementary Fig. 3 for assignment of the INS spectra)[9].

In Fig. 2d, the INS spectrum shows an additional transition at 5.3 meV, corresponding to solid orthohydrogen[10,12]. This contribution is present in the INS spectra of bulk solid hydrogen at atmospheric pressure (Fig. 2f), i.e., some solid hydrogen remains in the cell despite the evacuation performed at 160 K for the 135 MPa loading. Figure 2g and h show that the peak position of normal H₂ under pressure, ca. 10 meV, is close to the position of the rattling modes (doublet contribution) of the enclathrated hydrogen. In other words, enclathrated hydrogen at 0.1 MPa and 5 K performs in a similar fashion to normal solid hydrogen under high-pressure conditions (240 MPa).

The splitting of the contribution at ca. 10 meV in two components evidences a certain degree of anisotropy of the center-of-mass potential, attributed to hydrogen enclathrated in cages. In addition, the anisotropy with respect to the orientation of the hydrogen molecules entrapped in cages gives rise to a splitting of the $J = 0 \rightarrow 1$ rotational transition (E ~14 meV) with a significant asymmetry in the low energy tail, compared to the peak of solid H₂. These spectra provide a clear fingerprint of the successful formation of H₂ clathrate hydrates in the confined cavities of the $D_2O$-PPAC at 135 MPa and

above. To confirm the reproducibility of these results, similar experiments were performed using freshly prepared $D_2O$-PPAC sample after pressurizing with 135 MPa hydrogen at two different temperature conditions, i.e. 280 K and 273 K (stabilization time, 90 min). These experiments (Supplementary Fig. 4) constitute clear proof of the need to have liquid water to promote the formation of hydrogen clathrates, i.e., hydrogen clathrates are exclusively nucleated at ca. 280 K. However, further growth cannot be excluded during the subsequent cooling step down to 160 K.

INS and neutron diffraction spectra are measured simultaneously in VISION. Figure 3 compares the diffraction patterns for the $D_2O$-PPAC sample at different hydrogen (H₂) pressures. These results unambiguously show that there is no clathrate formation for pressures of 100 MPa and below (hexagonal ice is the only phase present in the carbon cavities with peaks at d-spacing 3.48, 3.60, and 3.82 Å). At 135 MPa and above, hexagonal ice nearly vanishes, and a completely new neutron diffraction pattern appears with peaks at 2.80, 3.40, 3.80, 4.25, 4.92, and 5.20 Å. These peaks match with an sII crystal structure as reported by Mao et al. for the hydrogen clathrate hydrates[4]. The absence of significant residual peaks associated with hexagonal ice at 135 and 200 MPa demonstrates that the water–to–hydrate conversion under confinement conditions is very high (only a small residual contribution at 3.44 Å due to hexagonal ice can be appreciated at 135 MPa). Compared to the bulk system where nucleation and growth kinetics are very slow (usually takes days or weeks, except in small–scale experiments (e.g., DAC with a volume <10 μm³)), under confinement conditions, we can perform the nearly complete conversion, in a massive scale (*ca.* 1 cm³), within minutes (<10 min—Supplementary Fig. 2). At this point, it is important to highlight that either enclathrated hydrogen or solid hydrogen are hard to be identified using neutron diffraction due to the high incoherent cross-sectional area of the hydrogen molecule. Exclusively deuterated species (e.g., $D_2O$ in an sII structure) do so[10].

To further confirm the promoting effect of the confined environment, the thermal stability of the confined H₂ clathrates was evaluated using neutron diffraction (ND) analysis after pumping the sample cell down to 0.1 MPa at 5 K. Figure 4 shows the ND patterns for two representative samples at temperatures ranging from 5 K and up

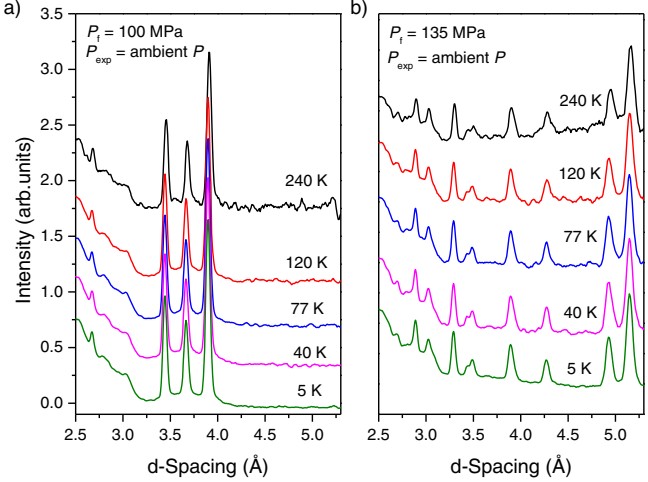

**Fig. 4 | Thermal stability studies of the confined crystals.** Thermal stability of confined (**a**) hexagonal ice and (**b**) $H_2$ clathrate hydrate up to 240 K followed by neutron diffraction. Before the experiment, sample $D_2O$-PPAC was pressurized at (**a**) 100 MPa and (**b**) 135 MPa. After pressurization, the samples were cooled to 5 K, pressure cell was decreased down to 0.1 MPa, and sample cell temperature was increased stepwise. Neutron diffraction studies demonstrate the high stability of the confined crystals, either hexagonal ice crystals (for sample $D_2O$-PPAC pressurized at 100 MPa) or sII hydrogen hydrate crystals (for sample $D_2O$-PPAC pressurized at 135 MPa). Source data are provided as a Source Data file.

to 240 K. As described above, 100 MPa is insufficient pressure to stabilize the $H_2$ clathrate. Under these conditions, hexagonal ice is the only phase present in the inner cavities of the activated carbon. Figure 4a shows that the confined ice crystals are fully stable at least up to 240 K[8]. A similar experiment in the sample pressurized at 135 MPa is shown in Fig. 4b.

Figure 4b clearly shows that the sII structure of the hydrogen clathrate hydrate is fully stable up to 240 K under atmospheric pressure conditions. This result is notable and proves the stabilizing effect of nanoconfinement in the preservation of crystal structures outside their stabilization zone. Previous studies from Mao et al. showed that the stability of the sII structure in bulk $H_2$ clathrates is limited to 145 K[4]. Similar thermal stability was described by Lokshin et al. for bulk $D_2$ clathrate hydrates, up to 163 K at 0.1 MPa[13]. Supplementary Fig. 5 shows the temperature evolution of the INS spectra corresponding to Fig. 4b. The hydrogen features are still observed at the highest temperature, confirming that the clathrate structure survives and that the hydrogen molecules are still enclathrated at 0.1 MPa of pressure. Furthermore, the disappearance of the contribution at 5.3 meV upon warming to 40 K confirms that this peak corresponds exclusively to solid hydrogen non-participating in the clathrate structure, in close agreement with observations in Fig. 2.

## Discussion

Overall, these results constitute the first experimental proof of an almost complete conversion of water into hydrogen clathrate hydrates in a massive scale and with fast nucleation and growth kinetics starting from pure liquid water. Confinement effects in the cavities of the PPAC sample promote the nearly complete conversion of $D_2O$ into clathrate crystals in $D_2O$-PPAC. Another striking finding is that confinement effects decrease the threshold pressure by more than 65 MPa (~30%) without incorporating additives, i.e., confined $H_2$ hydrates are formed at 135 MPa versus 200-220 MPa needed for the bulk system. Florusee et al. demonstrated that it is possible to stabilize clusters of $H_2$ clathrate hydrate at lower pressures using another approach, i.e., by incorporating a second guest component, THF, occupying the large cages[14]. These binary clathrate hydrates could be synthesized at

pressures around 5 MPa and 280 K. However, this approach has significant limitations. Indeed, the partial occupation of the large cages by THF limited the total amount of hydrogen stored (*ca.* 2.1 wt.% $H_2$)[14,15]. Smaller concentrations of THF (0.1 mol%) allowed to place $H_2$ molecules in the large cages at 12 MPa, the gravimetric capacity rising to ca. 4 wt.%[15]. However, the reaction was too slow to make it practical. In our approach, confinement at the nanoscale promotes the clathrate formation process under milder conditions than the same process in a non-confined environment, even when using pure liquid water[16–18]. Confinement effects are also reflected in the improved thermal stability of $H_2$ hydrates outside the normal stability zone. This enhanced stability (≥240 K vs 145 K) constitutes a step towards potentially applying these confined crystals as hydrogen reservoirs. The promoting effect of the 3D carbon network in the hydrogen clathrate formation process must be attributed to (i) the combination of a proper porous structure (a high surface area and the combined presence of micropores and mesopores) and (ii) a proper wettability ($D_2O$ must wet the carbon surface), but associated with the specific hydrophobic character needed to promote water-hydrogen interactions[19,20]. On one side, the surface of the activated carbon favors the proper orientation of the interfacial water molecules (tetrahedral ordering) to provide nucleation sites[21]. Additionally, confinement effects enhance hydrogen solubility in the confined water (increased gas density at the hydrophobic solid-water interface)[21–23]. These two characteristics and the extended water-gas interface in confined environments (Supplementary Fig. 6) explain the promoting role of our high-surface-area activated carbon material. Overall, carbon materials do not alter or modify the thermodynamics of the nucleation process itself but infer drastic changes in the nucleation and growth kinetics (decreased activation energy), thus speeding up the nucleation process at much lower pressures than the bulk system.

To conclude, the hydrogen storage capacity of the confined hydrates has been compared to the bulk system. Previous studies combining high-pressure Raman, infrared, x-ray, and neutron diffraction have proved the presence of multiple cage occupations in hydrogen hydrates, with four $H_2$ molecules in the large cages ($5^{12}6^4$) and two hydrogen molecules in the small cages ($5^{12}$)[4]. Under these conditions, the stoichiometry of the sII structure is 64 $H_2$·136 $H_2O$. Assuming the exact stoichiometry for the sII structure in bulk and confined nanospace, the gravimetric storage capacity of the bulk hydrate will be close to 5.0 wt.% (supplementary Table 2). In comparison, the capacity of the confined system will be slightly lower (≈ 4.1 wt.%), due to the additional weight of the carbon network. However, the main achievement of confined hydrates takes place in the volumetric storage capacity. Due to the presence of a strong adsorption potential and the extraordinary textural properties ($S_{BET}$ above 3500 $m^2$/g), sample PPAC can adsorb 4.1 times its weight in $H_2O$ without large changes in the external volume (water is mainly located in the inner cavities and in the interparticle space). Under these conditions (considering that the skeleton density of the PPAC sample is 1.90 g/ $cm^3$, and assuming the same density for the confined crystals as the bulk clathrates (0.94 g/ml)), the volumetric storage capacity is slightly lower than the bulk system (ca. 41.9 g/L) and significantly above the actual DoE target (30 g/L)[6]. These findings open the gate towards applying confined clathrate hydrates as hydrogen storage media for mobile or stationary applications[24]. However, these values have to be considered with caution since we are assuming the same stoichiometry for the bulk and the confined hydrogen clathrates based on the well-defined sII structure obtained with neutron diffraction (Fig. 3). Previous studies described in the literature using porous carbon materials have shown that this assumption is true for methane clathrates[7]. At this stage, more research is needed to modify the carbon network and/or to design novel porous materials with the proper porous structure and surface chemistry to promote the nucleation and growth of the confined hydrogen clathrates below 100 MPa.

## Methods

### Synthesis of the PPAC sample

Activated carbon PPAC was prepared from a petroleum residue (vacuum residue - VR) using KOH as activating agent. Before the activation step, the mesophase pitch was submitted to a pyrolysis treatment at 733 K for 1.5 h using 1 MPa $N_2$. After pyrolysis the pitch contains 93% of mesophase. In a subsequent step, activation was performed with KOH (KOH/pitch ratio 6:1) at 1073 K for 2 h using a nitrogen flow of 100 ml/min. Upon activation, the resulting material was washed with HCl (10 wt.%) and distilled water until neutral pH. In the final step, the synthesized carbon was dried overnight at 373 K.

### Sample characterization

Textural properties of the synthesized PPAC carbon were evaluated using nitrogen adsorption at cryogenic temperatures (77 K). Gas adsorption measurements were performed in a home-made mano-metric equipment designed and constructed by the LMA group. Nitrogen adsorption data were used to estimate the BET surface area, the micropore volume (after application of the Dubinin-Radushkevich equation), the mesopore volume and the total pore volume. Before the adsorption experiment, PPAC sample was out-gassed at 523 K for 4 h. The macroporosity was quantified by Hg porosimetry using a Poremaster-60 GT equipment from Quanta-chrome Instruments. The morphology of the synthesized activated carbon was analyzed by TEM (JEOL JEM-2100 F microscope) and FESEM (Merlin VP Compact system from ZEISS with a resolution of 0.8 nm at 15 kV and 1.6 nm at 1 kV). X-ray photoelectron analysis (XPS) were performed in a K-ALPHA Thermo Scientific equipment. XPS spectra were collected using an Al-K radiation (1486.6 eV), mono-chromatized by a twin crystal monochromator, yielding focused X-ray spot elliptical shaped with a major axis length of 400 μm at 3 mA x 12 kV. The alpha hemispherical analyzer was operated at the constant energy mode with survey scan pass energy of 200 eV to measure the whole energy band and 50 eV in a narrow scan to selectively measure the desired elements. The $CH_x$ in carbon 1 s score level was used as reference binding energy (284.6 eV).

### INS and ND experiments

Inelastic neutron scattering (INS) and neutron diffraction (ND) experiments were performed at the VISION beamline [https://neutrons.ornl.gov/vision] of the Spallation Neutron Source (SNS), Oak Ridge National Laboratory (ORNL). A closed cycle refrigerator cryostat was used to control the temperature of the sample in a range of 5-300 K. A Cu-Be high-pressure cell was used in combination with a manual piston compressor to deliver gas pressure in the range 0-200 MPa. Neutron powder diffraction data were collected simultaneously using the 90° detector bank. The CuBe cell does not produce Bragg peaks at d-spacing higher than 2.5 Å. Before the INS experiments, sample PPAC was impregnated dropwise with ultrapure $D_2O$ up to oversaturation (4.1 $g_{D2O}/g_{AC}$). Around 1 g of $D_2O$-PPAC was loaded to the Cu-Be cell and submitted to the temperature and pressure profile specified in Supplementary Fig. 2. All the reported INS experiments were performed only once. In the specific case of 135 MPa hydrogen pressure, two additional experiments were performed under iso-thermal conditions (stabilization step at 280 K and 273 K, respectively, for 90 min). After the isothermal step, samples were cooled down following the temperature and pressure profiles described in Supple-mentary Fig. 2.

## Data availability

Source data are provided with this paper. The data generated in this study have been deposited in the University of Alicante (UA) database at http://hdl.handle.net/10045/126860. Source data file is open access. Source data are provided with this paper.

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

## Acknowledgements

We would like to acknowledge financial support from Ministerio de Ciencia e Innovación (Project PID2019-108453GB-C21), MCIN/AEI/10.13039/501100011033 and EU "NextGeneration/PRTR" (Project PCI2020-111968 /3D-Photocat) – JSA. Neutron scattering experiments were performed at ORNL's Spallation Neutron Source, IPTS-27062, supported by the Scientific User Facilities Division, Office of Basic Energy Sciences, US DOE, under Contract No. DE-AC0500OR22725 with UT Battelle, LLC—J.S.A., Y.Q.C., L.D., A.J.R.C.. We gratefully acknowledge research support from the Hydrogen Materials—Advanced Research Consortium (HyMARC), established as part of the Energy Materials Network under the U.S. Department of Energy, Office of Energy Efficiency and Renewable Energy, Hydrogen and Fuel Cell Technology Office, under Contract Number DE-AC05-00OR22725—R.B.-X. This manuscript has been authored in part by UT-Battelle, LLC, under contract DE-AC05-00OR22725 with the US Department of Energy (DOE). The publisher acknowledges the US government license to provide public access under the DOE Public Access Plan (http://energy.gov/downloads/doe-public-access-plan).

## Author contributions

J.F.P. & C.C.C. performed the synthesis and characterization of the activated carbon. M.M.E. & J.S.A. participated in the evaluation and discussion of the characterization results. R.B.-X., Y.Q.C., & L.D. performed the INS experiments. A.J.R.C. participated in the evaluation and discussion of the INS data. A.J.R.C. & J.S.A. were the PIs of the project and participated in the design of the experiments, and writing the manuscript.

## Competing interests
