## [Peer Review File · Nature Communications]

REVIEWER COMMENTS

Reviewer #1 (Remarks to the Author):

The manuscript entitled „Rapid and efficient hydrogen clathrate formation in confined nanospace“ written by Farrando-Perez et al. describes the formation of hydrogen hydrates in high-surface area carbon materials. The authors used Petroleum Pitch Activated Carbon (PPAC) which was dropwise impregnated with D2O before pressurization with H₂. After pressurization with H₂ at defined pressures (1.0 kbar, 1.35 kbar and 2.0 kbar) the sample was cooled below the H₂ hydrate formation temperature (273-280 K). At 160 K the excess H₂ was released and the sample cooled to 5 K before inelastic neutron scattering and neutron diffraction was applied. The spectra confirmed the formation of H₂ hydrates at initial pressures of 1.35 kbar and above. The manuscript is well written and the results could possibly make an important contribution to the formation of hydrogen hydrates. Nevertheless, some questions arising from the reading of the manuscript remain unanswered to me. Before the manuscript is published, the following issues should be addressed:

I could not find any information on how many times the experiments on the formation of the H₂ hydrate and also on the decomposition were repeated at the respective pressure conditions. Would you be so kind as to add this information?

In this context I would be interested to know if the formation of solid orthohydrogen (signal at 5.3meV in the INS spectrum) has been observed several times and what could be the possible cause.

The macroscopically observed higher thermal stability of the H₂ hydrate phase in the pores of the activated carbon material could also only appear to be so. Would it not also be conceivable that the hydrogen released during the decomposition of the hydrate cannot escape from the nanopores, so that an increased pressure builds up in the pores, which delays further decomposition?

The authors expect the H₂ hydrate to form as soon as the temperature falls below 280 K and before the temperature falls below 273 K. They deduce this from further experiments, in which D₂O was pressed with H₂ once at 280 K and once at 273 K, and H₂ hydrate was formed exclusively at 280 K. I agree with the conclusion that this experiment shows that apparently no hydrate nucleation occurs when solid D₂O is pressurized. However, this does not preclude the possibility that a hydrate nucleation and growth process started at 280 K may continue at temperatures below 273 K. The formation of hydrates is an exothermic process and the heat of formation generated could sustain the process despite continuous cooling.

I also wonder if such rapid H₂ hydrate formation as suggested by the authors should not also appear as an exothermic event in the temperature curve in Fig S3? By the way, maybe you could complete the axis labeling at Fig S3 (T [K]).

It is also not entirely clear to me where the authors derive the H₂ capacity of their H₂ hydrate. I cannot see from which of the data shown this can be derived. The data published by e.g. Mao et al. showing that H₂ hydrate under the formation conditions chosen there included four H₂ molecules in the hexakaidecahedrons and two H₂ molecules in the pentagonal dodecahedron cannot be simply transferred. It is well known that both the chosen pressure conditions and the rate of formation have an influence on the cage occupancy, so that this way of calculating the H₂ capacity is quite speculative without direct experimental evidence. Therefore, I would recommend to remove this part of the text or at least point out that this calculation is quite speculative.

Reviewer #2 (Remarks to the Author):

This study established an effective route to form massive amount of hydrogen hydrate (sII) using

activated carbon materials, which may enhance the stability of sII and reduce the activation energy of the reaction. The experimental evidence for the completeness of the reaction, derived from inelastic neutron scattering and neutron diffraction, is enough strong that this remarkable establishment would be worth publishing from Nature Communications. Only a few minor concerns listed below should be addressed before the potential publication.

1. (In P5 line 17) The INS spectra may show some bulk solid hydrogen, which remains in the cell. However, no clear diffraction peaks from the solid hydrogen were observed. Because the INS intensities are generally so small compared to elastic scattering (diffraction), neutron diffraction should detect this solid hydrogen if exist. In that sense, the results from INS and ND seem contradicted.

2. This study claims PPAC play a role in both stabilizing sII and promoting the reaction kinetics, i.e., affects both chemical potentials and ΔG^* (activation energy of the reaction). The reason for changing chemical potentials may be the result of 'nano-confinement' as described in the discussion section, but no reason for reducing activation energy may not be written anywhere in this manuscript. It is well known in the reaction between water and hydrate that hydrates form on the surface of the water at the beginning of the reaction, and the formed hydrates disturb the further reaction because hydrogen could not penetrate hydrates on the surface and could not reach to water. This would be more familiar as the self-preservation effect in the transformation from methane hydrates to ice and methane. The reason for the fast kinetics should be given in the discussion section.

3. SI units may be preferable, so 'kbar' should be replaced with 'MPa'. '10⁻³ kbar' may be just written as ambient pressure (written so in the original paper by Lokshin et al.).

4. 'hexagonal ice' should be more clearly written as 'ice Ih', since there is another hexagonal ice, ice XVII (P6122).

Reviewer #3 (Remarks to the Author):

This manuscript reports the formation of hydrogen clathrate hydrate using a carbon material as the catalyst. The kinetics of clathrate hydrate formation are notoriously slow. This work therefore represents a major advance in this area of research. The experiments have been well designed making use of both neutron spectroscopy to probe the immobilisation of hydrogen and neutron diffraction to detect the formation of clathrate hydrates. I think the work should be published in Nat. Comm. once the authors have addressed the following points:

* line 43: Discuss how high the pressures need to be without the carbon material giving actual pressure values. In this context, the manuscript would benefit substantially from a blank experiment without the carbon material. Without this, it is hard for the reader to judge how well the carbon material actually works in facilitating the clathrate hydrate formation (which is the key point of the paper).

* Figure 1: For the benefit of the reader, it would be good to include the TEM image from the SI in the main figure. Perhaps SEM would give a more useful image than TEM?

* Figure 2: What is the origin of the Bragg features in the 2.75 to 3.1 Å range? The "h" in ice Ih needs to be italicised. Replace "A" with "Å" here and elsewhere.

* Figure 4: Can the authors show the decomposition of the clathrate hydrate in the right column?

* As a general comment, all the figures need to be improved. The figures should be understandable without having to read the caption in detail. For example, it should be clear what the circle in Figure 1 means etc.

* The manuscript would benefit from some sort of schematic illustration of the improved process.

* Based on their data, can the authors derive any quantitative information regarding the kinetics such as the lowering of the activation energy?

* Can the authors exclude the possibility that the carbon material binds some hydrogen as well? It is a highly porous material after all and the storage of hydrogen in carbon nanomaterials has been shown.

Answer to reviewers

First, we thank all the reviewers for their fruitful comments on our manuscript "*Rapid and efficient hydrogen clathrate formation in confined nanospace.*" We have addressed all the reviewers' comments, as shown below, and we have modified the manuscript accordingly.

Answer to Reviewer #1:

The manuscript entitled "Rapid and efficient hydrogen clathrate formation in confined nanospace" written by Farrando-Perez et al. describes the formation of hydrogen hydrates in high-surface area carbon materials. The authors used Petroleum Pitch Activated Carbon (PPAC) which was dropwise impregnated with D₂O before pressurization with H₂. After pressurization with H₂ at defined pressures (1.0 kbar, 1.35 kbar and 2.0 kbar) the sample was cooled below the H₂ hydrate formation temperature (273-280 K). At 160 K the excess H₂ was released and the sample cooled to 5 K before inelastic neutron scattering and neutron diffraction was applied. The spectra confirmed the formation of H₂ hydrates at initial pressures of 1.35 kbar and above. The manuscript is well written and the results could possibly make an important contribution to the formation of hydrogen hydrates. Nevertheless, some questions arising from the reading of the manuscript remain unanswered to me. Before the manuscript is published, the following issues should be addressed:

Answer: First, we would like to thank the reviewer for the nice words about our manuscript.

I could not find any information on how many times the experiments on the formation of the H₂ hydrate and also on the decomposition were repeated at the respective pressure conditions. Would you be so kind as to add this information?

Answer: The majority of the experiments were repeated only once. Due to time restrictions at the neutron facilities, we could not repeat all the experiments. However, the measurement at 1.35 kbar was repeated three times, using the conventional protocol described in Figure S2 and two additional experiments with specific stops at 270K and 280K for 90 minutes. These two additional experiments confirmed our finding that 1.35 kbar and temperatures above 273K are needed to successfully form the hydrogen clathrate.

This information was added to the experimental section (Page 11):

Around 1g of D₂O-PPAC was loaded to the Cu-Be cell and submitted to the temperature and pressure profile specified in fig S3. All the reported INS experiments were performed only once.

In this context I would be interested to know if the formation of solid orthohydrogen (signal at 5.3meV in the INS spectrum) has been observed several times and what could be the possible cause.

Answer: As explained in the manuscript, the contribution at 5.3 meV is due to solid hydrogen. We only observed this signal in one of the experiments (after pressurizing the sample holder with 1.35 kbar) but not for the other pressures tested. The main explanation for this signal is that some hydrogen must be left in the reaction chamber during the cooling process. This is the reason why we always included in the experimental protocol a pumping step at 160K, to release excess hydrogen, and to minimize the risk of hydrogen condensation. However, since the cooling step is always fast, sometimes it is not easy to remove all the non-adsorbed or non-enclathrated hydrogen. The confirmation of the nature of this peak (solid hydrogen non-participating in the clathrate) can be appreciated in Figure S6. Upon warming the sample to 40K, the signal at 5.3 meV disappears while the clathrate peaks remain unchanged.

We have added a sentence on page 8 to clarify the nature of the 5.3 meV signal:

.....of pressure. Furthermore, the disappearance of the contribution at 5.3 meV upon warming to 40K confirms that this peak corresponds exclusively to solid hydrogen non-participating in the clathrate structure, in close agreement with observations in Fig. 2.

The macroscopically observed higher thermal stability of the H₂ hydrate phase in the pores of the activated carbon material could also only appear to be so. Would it not also be conceivable that the hydrogen released during the decomposition of the hydrate cannot escape from the nanopores, so that an increased pressure builds up in the pores, which delays further decomposition?

Answer: As shown in the description of the activated carbon material, the porous network is mainly constituted of micropores ($V_{\text{micro}} \approx 1.06 \text{ cm}^3/\text{g}$), mesopores ($V_{\text{meso}} \approx 1.90 \text{ cm}^3/\text{g}$), and macropores. From our previous experience with gas hydrates and considering that the PPAC sample is oversaturated with D₂O, we believe that hydrogen clathrates must be formed preferentially in these three types of cavities, i.e. wide micropores, mesopores and macropores (the situation in small micropores is more complex due to the restricted space below the unit cell of the clathrate). Considering that micropores are connected with the external surface through the mesopores and macropores, it is easy to believe that the decomposition of the confined hydrates must proceed from large macro and mesopores to inner micropores. Based on this assumption, it is difficult to believe that a potential increase in pressure within the macro/mesocavities can delay or block the decomposition of the more internal gas hydrates. In our opinion, the improved stability is exclusively due to the nanoconfinement effects in the inner cavities.

The authors expect the H₂ hydrate to form as soon as the temperature falls below 280 K and before the temperature falls below 273 K. They deduce this from further experiments, in which D₂O was pressed with H₂ once at 280 K and once at 273 K, and H₂ hydrate was formed exclusively at 280 K. I agree with the conclusion that this experiment shows that apparently no hydrate nucleation occurs when solid D₂O is pressurized. However, this does not preclude the possibility that a hydrate nucleation and growth process started at 280 K may continue at temperatures below 273 K. The formation of hydrates is an exothermic process and the heat of formation generated could sustain the process despite continuous cooling.

Answer: We fully agree with the reviewer. With this experiment at two different temperatures (273K and 280K) we wanted to prove that liquid water is needed to promote the nucleation and growth of hydrogen clathrate. However, we cannot exclude that the growth of the already nucleated sites could continue at temperatures below 273K. Down to 160K, when excess hydrogen pressure is released, there is always the possibility that the gas hydrate growth continues, although with slower kinetics due to the low temperature.

To clarify this point, in page 6, we have added the following sentence:

These experiments (fig. S5) constitute clear proof of the need to have liquid water to promote the formation of hydrogen clathrates, i.e., hydrogen clathrates are exclusively nucleated at ca. 280K. However, further growth cannot be excluded during the subsequent cooling step down to 160K.

I also wonder if such rapid H₂ hydrate formation as suggested by the authors should not also appear as an exothermic event in the temperature curve in Fig S3?

Answer: We would expect a well-defined exothermic event for materials with uniform pores when all gas hydrates are formed at the same pressure or temperature conditions (see for instance the methane hydrate formation in MOFs: J. Am. Chem. Soc. 142 (2020) 13391). However, in the case of sample PPAC, the presence of a wide pore size distribution (including micropores, mesopores and macropores) widens the pressure window for the nucleation and growth due to the different environments, i.e. macro/mesopores are the first ones to participate in the nucleation process while inner micropores require more stringent conditions (Nature Communications 6 (2015) 6432). Based on this assumption and considering the wide pore size distribution for sample PPAC, one would expect a progressive crystal growth (from macro/meso to micropores) rather than a sudden step. In addition, one has to keep in mind that the CuBe cell is very thick to withstand high pressure, thus minimizing the thermal sensibility of the system.

By the way, maybe you could complete the axis labeling at Fig S3 (T [K]).

Answer: The label at Figure S3 has been modified accordingly.

It is also not entirely clear to me where the authors derive the H₂ capacity of their H₂ hydrate. I cannot see from which of the data shown this can be derived. The data published by e.g. Mao et al. showing that H₂ hydrate under the formation conditions chosen there included four H₂ molecules in the hexakaidecahedrons and two H₂ molecules in the pentagonal dodecahedron cannot be simply transferred. It is well known that both the chosen pressure conditions and the rate of formation have an influence on the cage occupancy, so that this way of calculating the H₂ capacity is quite speculative without direct experimental evidence. Therefore, I would recommend to remove this part of the text or at least point out that this calculation is quite speculative.

Answer: We fully agree with the reviewer that the assumption of the hydrogen occupancy in the synthesized hydrogen clathrates is pure speculation. Unfortunately, we could not estimate the amount of hydrogen forming clathrate at the high pressures of the experiments performed at ORNL. Under these conditions, pressure changes are small (compared to the base pressure of the chamber, always above 1 kbar). However, it is also true that the neutron diffraction pattern clearly shows the formation of the sII structure. From our previous experience with methane clathrates, the formation of the sI structure was always a guarantee that the stoichiometry was close to 5.75 H₂O per CH₄ molecule (stoichiometry of the bulk system), including the hydrates formed in the confined nanospace of porous materials. Considering this know-how, we are confident that the well-defined sII structure in the neutron diffraction spectra guarantees that the stoichiometry must be rather close to the one proposed for the bulk system, i.e., 64H₂·136H₂O. As we prefer to keep these values in the manuscript to give a general overview of the potential of these confined crystals (using an optimal stoichiometry) to compete with other systems (e.g., physisorption in MOFs) for hydrogen storage, we have modified the text to highlight the concerns raised by the reviewer.

In the original manuscript we already emphasized the uncertainty of these values:

Assuming the same stoichiometry for the sII structure in the bulk and in confined nanospace, the gravimetric storage capacity of the bulk hydrate will be close to 5.0 wt.% (table S2), while the capacity of the confined system will be slightly lower (\approx 4.1 wt.%), due to the additional weight of the carbon network.

But to make it clearer we have added the following paragraph in Page 10

However, these values have to be considered with caution since we are assuming the same stoichiometry for the bulk and the confined hydrogen clathrates based on the well-defined sII structure obtained with neutron diffraction. Previous studies described in the literature using porous carbon materials have shown that this assumption is true for methane clathrates (7). At this stage, more research is needed to modify the carbon network and/or to design novel porous materials....

Answer to Reviewer #2:

This study established an effective route to form massive amount of hydrogen hydrate (sII) using activated carbon materials, which may enhance the stability of sII and reduce the activation energy of the reaction. The experimental evidence for the completeness of the reaction, derived from inelastic neutron scattering and neutron diffraction, is enough strong that this remarkable establishment would be worth publishing from Nature Communications. Only a few minor concerns listed below should be addressed before the potential publication.

Answer: First, we would like to thank the reviewer for the nice words about our manuscript.

1. (In P5 line 17) The INS spectra may show some bulk solid hydrogen, which remains in the cell. However, no clear diffraction peaks from the solid hydrogen were observed. Because the INS intensities are generally so small compared to elastic scattering (diffraction), neutron diffraction should detect this solid hydrogen if exist. In that sense, the results from INS and ND seem contradicted.

Answer: This is an excellent point raised by the reviewer. To answer this question, one has to keep in mind that this study was performed combining D₂O and H₂. The reason behind this combination is the following:

Hydrogen atoms have a very high incoherent cross-sectional area and they cannot be detected using neutron diffraction studies. Only deuterated hydrogen will give a signal in the diffraction pattern. This is the reason why we cannot observe solid hydrogen in Figure 3.

Furthermore, this is also the reason why in the diffraction pattern we can identify the clathrate structure (since we use D₂O), but we cannot identify the enclathrated hydrogen. These details have been added to the manuscript.

In order to highlight this point raised by the reviewer, we have added the following paragraph to page 6:

At this point it is important to highlight that neither enclathrated hydrogen nor solid hydrogen can be identified using neutron diffraction due to the high incoherent cross-sectional area of the hydrogen molecule. Exclusively deuterated species (e.g., D₂O in an sII structure) do so (10).

2. This study claims PPAC play a role in both stabilizing sII and promoting the reaction kinetics, i.e., affects both chemical potentials and ΔG^* (activation energy of the reaction). The reason for changing chemical potentials may be the result of 'nano-confinement' as described in the discussion section, but no reason for reducing activation energy may not be written anywhere in this manuscript. It is well known in the reaction between water and hydrate that hydrates form on the surface of the water at the beginning of the reaction, and the formed hydrates disturb the further reaction because hydrogen could not penetrate hydrates on the surface and could not reach to water. This would be more familiar as the self-preservation effect in the transformation from methane hydrates to ice and methane. The reason for the fast kinetics should be given in the discussion section.

Answer: We would like to thank the reviewer for raising this interesting point. The presence of carbon in the synthesis media modifies the kinetics of the nucleation process. This effect must be attributed to the characteristics of the carbon material (surface chemistry and porous structure) and the enhanced water-liquid interface. To clarify the promoting effect of the carbon network, we have added in the supporting information an illustrative scheme of the benefits of confined hydrates vs. bulk hydrates. Additionally, we have addressed this discussion in the manuscript (Page 9).

On one side, the surface of the activated carbon favors the proper orientation of the interfacial water molecules (tetrahedral ordering) to provide nucleation sites (21). Additionally, confinement effects enhance hydrogen solubility in the confined water (increased gas density at the hydrophobic solid-water interface) (21-23). These two characteristics and the extended water-gas interfaces in confined environments (fig. S6) explain the promoting role of our high-surface area activated carbon material. Overall, carbon materials do not alter or modify the thermodynamics of the nucleation process itself but infers drastic changes in the nucleation and growth kinetics (decreased activation energy), thus speeding up the nucleation process at much lower pressures than the bulk system.

3. SI units may be preferable, so 'kbar' should be replaced with 'MPa'. '10⁻³ kbar' may be just written as ambient pressure (written so in the original paper by Lokshin et al.).

Answer: We have modified the manuscript accordingly.

4. 'hexagonal ice' should be more clearly written as 'ice Ih', since there is another hexagonal ice, ice XVII (P6122).

Answer: We have modified the manuscript accordingly.

Answer to Reviewer #3:

This manuscript reports the formation of hydrogen clathrate hydrate using a carbon material as the catalyst. The kinetics of clathrate hydrate formation are notoriously slow. This work therefore represents a major advance in this area of research. The experiments have been well designed making use of both neutron spectroscopy to probe the immobilisation of hydrogen and neutron diffraction to detect the formation of clathrate hydrates. I think the work should be published in Nat. Comm. once the authors have addressed the following points:

Answer: First, we would like to thank the reviewer for the nice words about our manuscript.

* line 43: Discuss how high the pressures need to be without the carbon material giving actual pressure values. In this context, the manuscript would benefit substantially from a blank experiment without the carbon material. Without this, it is hard for the reader to judge how well the carbon material actually works in facilitating the clathrate hydrate formation (which is the key point of the paper).

Answer: To clarify the promoting role of the carbon material vs bulk hydrates, we have incorporated a new scheme in fig S6 to emphasize the advantages of the confined system. The proposed scheme is

Here we want to emphasize the critical role of the carbon surface to promote proper orientation of the water molecules and increasing the density of hydrogen. We have also addressed this discussion in the manuscript, page 9:

Indeed, the surface of the activated carbon favors the proper orientation of the interfacial water molecules (tetrahedral ordering) to provide nucleation sites (21). Additionally, confinement effects enhance hydrogen solubility in the confined water (increased gas density at the hydrophobic solid-water interface) (21-23). These two characteristics and the extended water-gas interfaces in confined environments (fig. S6) explain the promoting role of our high-surface area activated carbon material. Overall, carbon materials do not alter or modify the thermodynamics of the nucleation process itself but infers drastic changes in the nucleation and growth kinetics (decreased

activation energy), thus speeding up the nucleation process at much lower pressures than the bulk system.

Concerning the final point raised by the reviewer, we did not perform the experiment with pure water (in the absence of carbon) due to time restrictions at the synchrotron. One has to keep in mind that we are working with large reactors (ca. 1 cm³), and under these conditions, the homogenous nucleation of gas hydrates is very slow (on the orders of 10⁻¹¹ nuclei/cm³·s (J.Am.Chem.Soc. 2012, 134, 19544-19547). In addition, the nucleation and growth process will be restricted to the gas-water interface, with the associated difficulty in comparing with the PPAC confined hydrates. Lokshin et al. (Appl. Phys. Lett. 2006, 88, 131909) demonstrated that more than 20h are needed to get accurate diffraction data, using ice instead of liquid water. In the specific case of our experiment (with pure water), these numbers in terms of time scale will be much higher, the experiment becoming unpractical. However, it is well-documented in the literature (Mao et al. and other papers) that pressures close to 2.0 kbar are needed to promote the formation process in bulk water.

* Figure 1: For the benefit of the reader, it would be good to include the TEM image from the SI in the main figure. Perhaps SEM would give a more useful image than TEM?

Answer: Following the reviewer's advice, we have added FE-SEM images of the activated carbon material. FESEM and TEM images have been transferred to Figure 1.

* Figure 2: What is the origin of the Bragg features in the 2.75 to 3.1 Å range?

Answer: The doublet at 2.88 and 3.02 Å are coming from the sII structure of the hydrogen clathrate. This can be appreciated in the theoretical pattern of the sII clathrate structure at the bottom of Figure 3.

The "h" in ice Ih needs to be italicised. Replace "A" with "Å" here and elsewhere.

Answer: We have modified these two points following the reviewer's advice.

* Figure 4: Can the authors show the decomposition of the clathrate hydrate in the right column?

Answer: Indeed, Figure 4 does not reflect the decomposition of the ice (a) and the hydrogen clathrate (b) but rather the thermal stability of these confined crystals at ambient pressure. Figure 4 shows that both systems are thermally stable at atmospheric pressure up to 240 K. Unfortunately, we did not measure above this temperature. Still, this improved stability constitutes another goal of the confined crystals vs. bulk systems (in the bulk system, the decomposition temperature was 145K, as described by Mao et al.).

* As a general comment, all the figures need to be improved. The figures should be

understandable without having to read the caption in detail. For example, it should be clear what the circle in Figure 1 means etc.

Answer: We have improved the description of all the figures. See changes in yellow in the revised version of the manuscript.

* The manuscript would benefit from some sort of schematic illustration of the improved process.

Answer: We have added a new scheme in the supporting information (shown above in this document) to illustrate the promoting effect of our approach as compared to the bulk system. We added also a more detailed discussion of the formation mechanism and the advantages of activated carbon as a host structure.

* Based on their data, can the authors derive any quantitative information regarding the kinetics such as the lowering of the activation energy?

Answer: One of the main limitations working with these high-pressure values is the impossibility to accurately determining the amount of hydrogen trapped in the form of hydrogen clathrates. We know from our previous experience that the water-to-hydrate yield in these kinds of carbon materials is close to 100% for methane as a guest molecule (e.g., Nature Commun. 6 (2015) 6432), and we also know that this conversion process is fast (within minutes). However, these experiments are performed in conventional gas adsorption equipment, with the possibility to measure pressure changes accurately and, indirectly, to estimate compositional and kinetic parameters. However, the high pressures needed to grow these hydrogen clathrates require specific high-pressure systems not available in a conventional lab. Under these high-pressure conditions, it is impossible to accurately estimate the amount of hydrogen retained in the form of clathrate. As raised by the reviewer, the determination of the activation energy would be a possibility but we do not have enough experimental data to do so.

Anyhow, we have added a new paragraph in the discussion section (Page 9) to address this point:

Indeed, the surface of the activated carbon favors the proper orientation of the interfacial water molecules (tetrahedral ordering) to provide nucleation sites (21). Additionally, confinement effects enhance hydrogen solubility in the confined water (increased gas density at the hydrophobic solid-water interface) (21-23). These two characteristics and the extended water-gas interfaces in confined environments (fig. S6) explain the promoting role of our high-surface area activated carbon material. Overall, carbon materials do not alter or modify the thermodynamics of the nucleation process itself but infers drastic changes in the nucleation and growth kinetics (decreased activation energy), thus speeding up the nucleation process at much lower pressures than the bulk system.

* Can the authors exclude the possibility that the carbon material binds some hydrogen as well? It is a highly porous material after all and the storage of hydrogen in carbon nanomaterials has been shown.

Answer: This is a good point argued by the reviewer. When we perform the Inelastic Neutron experiments there is always the possibility that some hydrogen remains adsorbed in the inner porous structure of the activated carbon material. However, it is also true that in the experimental conditions used, i.e. after pumping excess hydrogen at 160K, it is very rare that some hydrogen will remain trapped in the structure. However, we cannot exclude this possibility and the best way to confirm this statement is the INS peak at 5.3 meV. Indeed, we had this problem when dosing the chamber with 1.35 kbar of hydrogen. As shown in Figure 2F, the contribution of solid hydrogen is clearly visible at 5.3 meV. This contribution reflects exactly this additional hydrogen trapped in the cavities of the carbon material as solid bulk hydrogen. However, this extra hydrogen does not modify and/or alter the parallel formation of hydrogen clathrates, i.e. it is non-enclathrated hydrogen. The best confirmation that this solid hydrogen is non-participating in the clathrate comes in figure S5. Upon warming to 40K, this peak disappears immediately whereas the hydrogen clathrate signal remains unchanged. This point has been further clarified in Page 8

Furthermore, the disappearance of the contribution at 5.3 meV upon warming to 40K confirms that this peak corresponds exclusively to solid hydrogen non-participating in the clathrate structure, in close agreement with observations in Fig. 2.

REVIEWERS' COMMENTS

Reviewer #1 (Remarks to the Author):

It is highly appreciated that the authors have taken up essential points of criticism and revised their manuscript accordingly. Nevertheless, there are the following two points that I do not yet find satisfactorily clarified.

1) I find it very difficult that the experiments were only done once. There is certainly no doubt that hydrogen hydrates form under the conditions mentioned, but it is difficult to make any statements about stability or formation kinetics beyond that, because the results were not reproduced. I certainly understand the difficulty that the experiments could not be repeated due to the limited measurement times. Nevertheless, I would recommend to be a little more careful when interpreting the data and to use the conjunctive rather than the empirical.

2) In connection with the stability of the hydrogen hydrates, another point also arises from the authors' answer to one of my questions. The authors write in their answer: "In addition, one has to consider that the CuBe cell is very thick to withstand the high pressure, which minimises the thermal sensitivity of the system". This led me to ask in what time sequence the heating of the sample took place or, in other words, how much time was given to the system between each temperature step to reach thermal equilibrium. Since the cell used apparently has poor thermal conductivity, there could be a danger here that if the heating curves are too fast, the temperature in the sample will be significantly lower than the temperature in the cryostat (and thus the temperature displayed). It would be helpful to take this aspect into account and provide appropriate information.

In my opinion, these points should be included in the revision of the paper before the manuscript is published.

Reviewer #2 (Remarks to the Author):

The modified manuscript reflect most of my previous comments, but I found a slight mistake in the point 1 about the effect of hydrogen on neutron diffraction.

In fact, hydrogen atom has comparable coherent scattering length to deuterium, so that diffraction from hydrogen or H-guest in sII hydrate if intensities are strong enough. Most of the case, the incoherent scattering hinders Bragg reflections if hydrogen contents are too large, so I would understand solid H₂ may not have sufficient intensity for observation. And I suppose Bragg intensities of sII are also altered due to the guest hydrogen. Thus, I suggest 'neither nor can be identified' in the added sentence may be 'either or ... are hard to be identified' and delete the followed sentence 'Exclusively deuterated species (e.g., ...)' since it is obvious.

Reviewer #3 (Remarks to the Author):

Dear Editor,

The authors have successfully addressed my concerns and I am therefore very happy to recommend the publication of the article.

Answer to reviewers

First, we thank all the reviewers for their fruitful comments on our manuscript "*Rapid and efficient hydrogen clathrate formation in confined nanospace.*" We have addressed all the reviewers' comments, as shown below, and we have modified the manuscript accordingly.

Answer to Reviewer #1:

It is highly appreciated that the authors have taken up essential points of criticism and revised their manuscript accordingly. Nevertheless, there are the following two points that I do not yet find satisfactorily clarified.

1) I find it very difficult that the experiments were only done once. There is certainly no doubt that hydrogen hydrates form under the conditions mentioned, but it is difficult to make any statements about stability or formation kinetics beyond that, because the results were not reproduced. I certainly understand the difficulty that the experiments could not be repeated due to the limited measurement times. Nevertheless, I would recommend to be a little more careful when interpreting the data and to use the conjective rather than the empirical.

Answer: We would like to thank the reviewer for the comments on the stability of the confined hydrogen clathrates. Maybe we did not explain properly this point in our first rebuttal. Initial experiments described in Figure 2 were performed only once due to the limited accessibility to the neutron facilities. However, we also had the same concerns of the reviewer. To answer this question, we added two additional experiments but using isothermal steps. More specifically, we applied two steps of 90 min at 273K and 280K. These two experiments were very useful to confirm the reproducibility of the experiments and to identify the necessity to have liquid water in order to promote the nucleation and growth of these hydrates.

In order to clarify these points, we have modified the text and the experimental section, accordingly:

Results

To confirm the reproducibility of these results, similar experiments were performed using freshly prepared D₂O-PPAC sample after pressurizing with 135 MPa hydrogen at two different temperature conditions, i.e. 273K and 280K (stabilization time, 90 min).

Experimental section

In the specific case of 135 MPa hydrogen pressure, two additional experiments were performed under isothermal conditions (stabilization step at 273K and 280K, respectively, for 90 min). After the isothermal step, samples were cooled down following the temperature and pressure profiles described in fig. S2.

2) In connection with the stability of the hydrogen hydrates, another point also arises from the authors' answer to one of my questions. The authors write in their answer: "In addition,

one has to consider that the CuBe cell is very thick to withstand the high pressure, which minimises the thermal sensitivity of the system". This led me to ask in what time sequence the heating of the sample took place or, in other words, how much time was given to the system between each temperature step to reach thermal equilibrium. Since the cell used apparently has poor thermal conductivity, there could be a danger here that if the heating curves are too fast, the temperature in the sample will be significantly lower than the temperature in the cryostat (and thus the temperature displayed). It would be helpful to take this aspect into account and provide appropriate information. In my opinion, these points should be included in the revision of the paper before the manuscript is published.

Answer: We would like to thank the reviewer for raising the concern about the temperature difference between the sample holder and the cryostat. In the experimental setup at VISION end-station, there are two thermocouples. One is connected to the sample cell and the second one is connected to the cryostat. We have continuous recording of both temperatures and the temperature difference is always 3-4K. To avoid uncertainty, the temperature reported in Fig. 4 and Fig. S5 corresponds to the sample cell temperature. This point has been clarified in these two figures.

Answer to Reviewer #2:

The modified manuscript reflect most of my previous comments, but I found a slight mistake in the point 1 about the effect of hydrogen on neutron diffraction. In fact, hydrogen atom has comparable coherent scattering length to deuterium, so that diffraction from hydrogen or H-guest in sII hydrate if intensities are strong enough. Most of the case, the incoherent scattering hinders Bragg reflections if hydrogen contents are too large, so I would understand solid H₂ may not have sufficient intensity for observation. And I suppose Bragg intensities of sII are also altered due to the guest hydrogen. Thus, I suggest 'neither nor can be identified' in the added sentence may be 'either or ... are hard to be identified' and delete the followed sentence 'Exclusively deuterated species (e.g., ...)' since it is obvious.

Answer: We fully agree with the reviewer. Incoherent back scattering of hydrogen is high and hinders Bragg reflections. Changes have been included.